# Stereotactic Ablative Radiotherapy Using CALYPSO^®^ Extracranial Tracking for Intrafractional Tumor Motion Management—A New Potential Local Treatment for Unresectable Locally Advanced Pancreatic Cancer? Results from a Retrospective Study

**DOI:** 10.3390/cancers14112688

**Published:** 2022-05-29

**Authors:** Hrvoje Kaučić, Domagoj Kosmina, Dragan Schwarz, Andreas Mack, Hrvoje Šobat, Adlan Čehobašić, Vanda Leipold, Iva Andrašek, Asmir Avdičević, Mihaela Mlinarić

**Affiliations:** 1Specijalna bolnica Radiochirurgia Zagreb, Ulica Dr. Franje Tuđmana 4, 10431 Sveta Nedelja, Croatia; domagoj.kosmina@radiochirurgia.hr (D.K.); dragan.schwarz@radiochirurgia.hr (D.S.); hrvoje.sobat@radiochirurgia.hr (H.Š.); adlan.cehobasic@radiochirurgia.hr (A.Č.); vanda.leipold@radiochirurgia.hr (V.L.); iva.andrasek@radiochirurgia.hr (I.A.); asmir.avdicevic@radiochirurgia.hr (A.A.); mihaela.mlinaric@radiochirurgia.hr (M.M.); 2Sveučilište Josipa Jurja Strossmayera u Osijeku—Medicinski Fakultet Osijek, Josipa Huttlera 4, 31000 Osijek, Croatia; 3Medicinski Fakultet Sveučilišta u Rijeci, Braće Branchetta 20/1, 51000 Rijeka, Croatia; 4Sveučilište Josipa Jurja Strossmayera u Osijeku—Fakultet za Dentalnu Medicinu i Zdravstvo Osijek, Crkvena Ulica 21, 31000 Osijek, Croatia; 5Swiss NeuroRadiosurgery Center, Bürglistrasse 29, 8002 Zürich, Switzerland; a.mack@snrc.ch

**Keywords:** Calypso^®^ Extracranial Tracking, fiducial-based tumor motion management, locally advanced pancreatic cancer, SABR, SBRT

## Abstract

**Simple Summary:**

An estimated 30–40% of patients with pancreatic cancer are at an unresectable locally advanced stage at the time of diagnosis, and this represents a particular problem in oncology due to the challenges in local treatment. The aim of this study was to investigate the potentially positive impact of local dose escalation during stereotactic ablative radiotherapy using intrafractional fiducial-based motion management on clinical outcomes. The system used for motion management in our study was Calypso^®^ Extracranial Tracking, and this is the first reported application of this system for locally advanced pancreatic cancer. Using very narrow safety margins around the lesion during the treatment, we were able to effectively spare surrounding healthy organs and safely apply median biological effective dose of 112.5 Gy. This approach, especially when combined with systemic therapy, resulted in very favourable one-year local tumor control of 100% and median overall survival of 24 months, with no grade > 2 toxicities.

**Abstract:**

(1) Background: The aim of this study was to evaluate the efficacy and safety of SABR for LAPC using Calypso^®^ Extracranial Tracking for intrafractional, fiducial-based motion management, to present this motion management technique, as there are yet no published data on usage of Calypso^®^ during SABR for LAPC, and to report on our clinical outcomes. (2) Methods: Fifty-four patients were treated with SABR in one, three, or five fractions, receiving median BED_10_ = 112.5 Gy. Thirty-eight patients received systemic treatment. End points were OS, FFLP, PFS, and toxicity. Actuarial survival analysis and univariate analysis were investigated. (3) Results: Median follow-up was 20 months. Median OS was 24 months. One-year FFLP and one-year OS were 100% and 90.7%, respectively. Median PFS was 18 months, and one-year PFS was 72.2%. Twenty-five patients (46.3%) were alive at the time of analysis, and both median FU and OS for this subgroup were 26 months. No acute/late toxicity > G2 was reported. (4) Conclusions: SABR for LAPC using Calypso^®^ presented as an effective and safe treatment and could be a promising local therapeutic option with very acceptable toxicity, either as a single treatment or in a multimodality regimen. Dose escalation to the tumor combined with systemic treatment could yield better clinical outcomes.

## 1. Introduction

Pancreatic cancer is currently the fourth leading cause of cancer death in Europe and the United States, and the seventh worldwide, and will potentially represent the second leading cause of cancer death by 2030 [1,2]. Ductal adenocarcinoma represents 85% of all pancreatic tumours [3].

In general, the five-year survival rate for patients with pancreatic cancer is less than 10% [4]. There are several reasons contributing to the overall poor prognosis for those patients, e.g., deep retroperitoneal location of the pancreas in the abdomen, leading to late symptoms and diagnosis, an aggressive biology with early metastasis (roughly in 50% of patients at presentation), and the presence of micrometastasis in apparently localized cases [4]. Furthermore, the disease ruins patients’ performance status dramatically, limiting their ability to withstand aggressive treatments, and is resistant to many antineoplastic drugs [5,6].

Only radical surgical resection with R0 margin significantly prolongs survival. However, a small portion of patients (<20%) are candidates for radical surgery, as the tumor is typically diagnosed at an advanced stage. Surgery for pancreatic cancer has significant postoperative morbidity, with rates ranging from 30% to 60%, but with improvements of surgical techniques and perioperative care, the mortality and morbidity rates have decreased in recent years. Major postoperative complications include pancreatic leak or fistula, intra-abdominal abscess, bile leak, postoperative haemorrhage, delayed gastric emptying, infection, and wound dehiscence [7,8,9,10,11]. Diabetes mellitus, temporary or permanent, is also one of the postoperative complications.

In recent years, stereotactic ablative body radiotherapy (SABR) has become an increasingly popular option as an alternative, adjuvant, and consolidation treatment for pancreatic cancer, as well as a re-treatment for the recurrent disease [12]. SABR delivers high doses in a few (typically 1–5) fractions, characterized with rapid dose fall-off outside the target volume [13,14], offering potential improvement of local tumour control [15,16,17,18], sparing of adjacent radiosensitive organs-at-risk (OARs) and consequently lowering the treatment toxicity [19,20]. Compared with conventionally fractionated radiotherapy, SABR has shown significantly improved OS for locally advanced disease [21]. Patients with unresectable, locally advanced pancreatic cancer (LAPC) represent a particular problem in oncology, due to the challenges in local treatment. LAPC is characterised with an absence of regional or distant disease on diagnostic imaging but is surgically unresectable due to the tumor’s extensive involvement of mesenteric and hepatic artery, mesenteric and portal vein and celiac axis [22].

The last Executive Summary of an ASTRO Clinical Practice Guideline [23] made comprehensive recommendations regarding the treatment of pancreatic cancer. The guidelines consider preoperative SABR for downstaging of LAPC to eventual surgery not appropriate, as SABR (with the total dose of 33–40 Gy in 6.6–8 Gy fractions) should follow the chemotherapy as a definitive treatment. The treatment volume should include only the gross tumor volume (GTV), with no elective nodal treatment. Respiratory, volumetric, or fiducial motion management and intensity modulated radiotherapy (IMRT) or volumetric arc therapy (VMAT) are recommended, as well as antiemetic prophylaxis. Patients should receive intravenous contrast at CT simulation.

Motion management plays a crucial role in sparing highly radiosensitive OARs adjacent to the pancreas (the duodenum, stomach, and small bowel), which represents the major limiting factor for dose escalation [24]. There are several motion management techniques. Respiratory mitigation using abdominal compression, respiratory gating, irradiation of 4D-CT generated internal target volume (ITV), and fiducial-based intrafractional motion tracking (typically used with robotic arm-based linacs) are used for patients in free breathing (FB). For irradiation in deep breath-hold (DBH), surface guidance or intrafractional tracking with cine magnetic resonance imaging (MRI) is used [23,24,25].

Regarding the motion management during SABR for pancreatic cancer, the recent ESTRO ACROP Guidelines for Target Volume Definitions in Pancreatic Cancer [25] recommended the individual motion detection and consideration during treatment planning with 4D-CT as the preferred and cine MRI as an alternative method [26,27,28], and irradiation in specific breathing phases using gating, active breathing coordinator, or real-time tumour tracking [29,30,31,32,33,34].

An estimated 30–40% of patients with pancreatic cancer are presented in a locally advanced stage at the time of diagnosis [35]. The goal of this study is to evaluate the efficacy and safety of SABR as a local treatment for patients diagnosed with LAPC, using the Calypso^®^ tracking system (Calypso^®^) for intrafractional, fiducial-based motion management. To our best knowledge, there are no published data on the usage of Calypso^®^ as an intrafractional motion management system during SABR for LAPC, although the system is commonly used for prostatic cancer [36,37], and FDA approved to improve the precision of radiotherapy and radiosurgery treatments for cancer [38]. The Calypso^®^ system uses continuous active tumor tracking during the treatment, as the tumor-implanted fiducial transponders are tracked electromagnetically by the array, in real time, with 20 Hz frequency and submillimetre accuracy, using non-ionizing radio frequencies. Each transponder (8.8 mm long with 1.85 mm in diameter) contains a capacitor and an inductor coil sealed in glass.

We did not present the analysis of the pancreas movements in current paper, as the evaluation of total pancreatic movements using the Calypso^®^ system was presented in a greater detail in our previously published paper [39]. We aimed primarily to report on our clinical outcomes of SABR using this motion management technique for LAPC.

## 2. Patients and Methods

### 2.1. Patients

Medical data of 63 patients diagnosed with LAPC that were treated between April/2017 and January/2021 in our institution were analysed. Nine patients were not available for regular follow up. The remaining 54 patients made themselves available for regular follow-up, thus being consecutively enrolled into in this retrospective, single-arm, and single-institution observational study, approved by the institutional ethics committee. Patients’ characteristics are shown in Table 1.

Prior to treatment, all patients were discussed and approved for SABR by our institution’s multidisciplinary tumour board, consisting of a radiation oncologist, a pancreatic/biliary surgeon, a radiologist, a medical physicist, and a medical oncologist. Inclusion criteria were: unresectable, histologically proven pancreatic adenocarcinoma, age ≥ 18, ECOG 0–2, negative regional lymph nodes with no signs of distant metastasis, gastric or duodenal obstruction on diagnostic imaging, and no previous abdominal radiotherapy. Unresectable pancreatic cancer was defined according to the arterial and venous criteria for resectability status using the recommendations from NCCN guidelines and the American Hepato–Pancreato–Biliary Association/Society of surgical Oncology/Society for Surgery of the Alimentary Tract [22,40]. All procedures performed were in accordance with the 1964 Helsinki Declaration (and its later amendments) or comparable ethical standards, as well as the national medical ethical standards. Signed informed consent was obtained from all subjects involved in the study.

As our institution does not apply systemic therapy, it was indicated in cooperation with, and provided by each patient’s referring medical oncologists’ team. Systemic therapy was held at least seven days before and after SABR, due to the possible toxicities of concurrent application.

### 2.2. Patients’ Preparations for the Treatment

The treatment in DBH was generally preferred, due to better visibility of the tumor and OARs on both planning MSCT and daily CBCTs in DBH, and faster dose delivery using VMAT. Prior to treatment planning, all patients were tested using Calypso^®^, if they could adequately (repeatedly and consistently for at least 20 s) hold their breath. If needed, further breathing coaching was performed, and the test was repeated. For patients that were not able to adequately keep the DBH by any means, treatment was performed in FB (Table 2.).

All patients were provided with written recommendations on diet and instructed to take proton pump inhibitors and antiflatulent drugs (e.g., simethicone), starting on the day of transponder implantation, to reduce flatulence and weight loss during the treatment, in order to minimize daily anatomical variations. For the same reasons, the time from the planning to the start of the treatment was kept as short as possible, typically up to 7 days. All patients received a combination of IV administered antiemetics and spasmolytics on a day of each fraction in our day clinic.

### 2.3. Calypso^®^ Tracking System

Calypso^®^ (Varian Medical Systems, Palo Alto, CA, USA) is a fiducial-based intrafractional motion management system, which is FDA approved for soft tissue tumour lesions motion management. The Calypso^®^ system consists of an electromagnetic array and tumour-implanted fiducials, Calypso Beacon^®^ transponders.

Transponders are 8.8 mm long with 1.85 mm in diameter, containing a capacitor and an inductor coil sealed in glass. The array, which is positioned above the patient during the treatment, detects the position and the translational and rotational movements of the transponders. The Calypso^®^ system provides continuous 3-dimensional intrafractional motion management of all possible tumor movements (both intrinsic and caused by the movements of a patient) in a real time, with 20 Hz frequency and submillimetre accuracy. The transponders, which are non-toxic and hypoallergenic, are designed for a percutaneous implantation into, or adjacent to the lesion. The special implantation needle is provided by the manufacturer. The system uses non-ionizing radio frequencies to localize the transponders. The Calypso^®^ system calculates by default the geometric center of the transponders (called the “centroid”) based on the detected initial location of each individual transponder. During SABR, Calypso^®^ actually tracks the centroid’s motions. Three implanted transponders are needed for the Calypso^®^ system to track centroid’s translational and rotational movements, and two implanted transponders are needed to track the centroid’s translational movements solely.

According to the manufacturer’s specifications, the lag time of the system is up to 100 milliseconds. Figure 1 shows the implantation needle, electromagnetic array, and Beacon^®^ transponders (the figure is taken from [39]).

Optimally three, or a minimum of two Beacon^®^ transponders were implanted percutaneously by a skilled and educated interventional radiologist from our institution using the equipment provided by the manufacturer, into or adjacent to the lesion. Implantation was performed under the CT-guidance using local anaesthesia, similarly to the pancreatic biopsy. According to the manufacturer’s manual, the distance between implanted transponders was 1–7 cm (minimal–maximal), to provide for an accurate motion tracking. To allow sufficient time for their in-site stabilization and to prevent possible migration, the transponders were implanted typically 10–14 days prior to treatment planning. All MR imaging for planning purposes was performed prior to transponder implantation, typically on the same day, as the transponders induce local artefacts on MR images. Every patient remained for 8 h in our institution after the implantation, due to the routine blood work for possible internal bleeding, and observation.

Contraindications for Beacon^®^ transponders implantation included: neuromuscular diseases, coagulopathies, acute infection disease and general contraindications for contrast enhanced CT scan. Unfavourable patient anatomy in the abdomen, if evaluated by the interventional radiologist, was also considered a contraindication. We noticed no complications or side effects during or after the fiducials’ implantation.

### 2.4. Stereotactic Ablative Radiotherapy

A contrast-free multi slice computed tomography (MSCT) scan in FB, DBH (inhale phase), and phase-gated 4D-CT study sets, with a slice thickness of 1 mm, and contrast-free MRI in DBH of the abdomen (T1 and T2 with high spatial fidelity) were routinely acquired for all patients.

Patients were treated either in FB or in a DBH, according to the criteria defined in the chapter “Patients’ preparations for the treatment”.

For the patients that were treated in FB:
The planning 4D-CT in a late exhale phase was used and coregistered with MRI.CTVs were delineated on the T1 or T2 images of the MRI, with further corrections (if needed) on phase-gated 4D-CT scans.


For the patients that were treated in DBH:
The planning MSCT in DBH was used and coregistered with MRI.CTVs were delineated on the T1 or T2 images of the MRI, with further corrections (if needed) on MSCT scans in DBH.


The clinical target volume (CTV) was defined as the gross tumor volume (GTV), with no additional margins. We followed the ESTRO, ASTRO and NCCN recommendations on target and OAR delineation.

The deformable registration methods were used as needed. If the lesion or the OARs were not clearly visible on a contrast-free imaging, additional contrast-enhanced MSCT (with late arterial phase) and/or contrast-enhanced MR of the abdomen were additionally acquired for coregistration with planning MSCT. As the MR sequences with contrast enhancement tend to overestimate the actual volumes, the contrast-free MR imaging was generally preferred.

All patients were in a supine position during SABR, on either a wing-board or vacuum pillow, with the arms positioned above the head. We used no further immobilization methods. A Varian EDGE^®^ linear accelerator (Varian Medical Systems, Palo Alto, CA, USA) was used for a treatment delivery. Treatments were conducted on consecutive days, with pauses over the weekends.

SABR plans were optimized and delivered using multiple coplanar arcs—VMAT, or multiple noncoplanar IMRT sliding window fields. Flattening filter free photon beams with energies of 6 MV and/or 10 MV and dose rates up to 1400 and 2400 MU/min, respectively, were used. Beam energies and dose delivery techniques were chosen to achieve the best dose distributions, while having plans with low modulation and high QA passing rates. Alpha/beta ratio = 10 Gy was used to calculate the Biological effective dose to tumor (BED_10_). Fractionation regimens, corresponding BED_10_ and number of patients are presented in the Table 3. The optimal fractionation regimen was determined for each patient individually, to achieve the goal of optimal OARs sparing.

The dose was applied extremely heterogeneously. Typically, the mean dose to the PTV was pushed considerably higher than the prescription dose, and there were no planning constraints on the maximum dose as long as it was located inside the PTV. The result was a highly heterogeneous dose distribution inside the PTV, with an average maximum of 136.3% (ranging 129.6 to 143.2%) of the prescription dose. The optimization of the dose distribution was performed with the purpose of achieving the following clinical goal: a required target coverage of V (98–99.5%) = 80% of the prescribed dose for the PTV (Figure 2.). PTV was generated using 3 mm margin to CTV for all patients. The PTV-CTV margin was calculated using van Herk’s formula (2.5Σ + 0.7σ → 2.5 × 0.6 mm + 0.7 × 2 mm = 2.9 mm) to estimate the systemic error (Σ = 0.6 mm, as determined by end-to-end tests), and the random error (σ = 2 mm, defined as gating windows), in X, Y and Z directions. The average conformity index was 1.09 (ranging 1.04 to 1.14), defined as the ratio of the volume of the 80% isodose line dose to the volume of the target (PTV).

The OARs were divided into two groups: the primary OARs, which are directly adjacent to pancreas and highly radiosensitive, and the other OARs. Primary OARs were the stomach, the duodenum, and the small intestine. For primary OARs, we used constraints according to Murphy et al. [41]. We followed AAMP recommendations for the dose-volume constraints for the other OARs [42]. The Table 4 summarizes the dose–volume constraints. Target coverage was prioritized over OAR sparing as long as stated OAR constraints were met.

Calypso^®^ was used as a beam on-off phase-gating technique for both patients treated in a DBH and in FB. During the treatment, the centroid was allowed to move within a gating window of 2 mm in any direction (lateral, longitudinal or vertical) from the treatment position before the beam was shut off. For DBH patients, VMAT was chosen as a dose delivery technique for its general faster performance and better dose distribution compared to IMRT. As opposed, for FB patients, the dose delivery was, in our opinion, significantly more favourable with IMRT. The reason for this was frequent moving of the transponders from the treatment position in FB, which could, due to the frequent “stops-and-goes” of the gantry, make the dose delivery with VMAT less reliable [43].

Initial positioning was performed in DBH or FB before each fraction as follows:For the patients treated in DBH we used planning MSCT in DBH for coregistration with cone beam CT (CBCT)For the patients treated in FB we used planning 4D-CT reconstructed in all breathing phases (“Average intensity projection”) for coregistration with CBCT.

We used soft tissue and bony anatomy and the transponders for registering, with weight given to the transponders’ position, and routinely repeated CBCT for every patient after the 50% of the dose was delivered for the fraction, to recheck any possible mismatch of CTV or OARs. Prior to beam on, the Calypso^®^ system initially checked that the actual position and relation of the transponders on a treatment table represented the planned transponder’s position and relation. A geometric deviation up to 2 mm (±1 mm in a single direction) and/or a rotation up to 20 degrees (±10 degrees in a single plane), according to manufacturer’s manual, were acceptable. If needed, further corrections were made using CBCT. After the treatment started, the beam was on as long as the centroid remained within the 2 mm gating windows in all directions. Moreover, as long as the centroid remained within the gating windows, it was assumed that there were no significant OAR movements.

### 2.5. Response Evaluation and Follow-Up

Follow up was regularly scheduled every three months after SABR by the treating radiation oncologist with clinical examination and a contrast-enhanced MSCT scan. Further diagnostic MRI imaging (contrast enhanced) was performed if MSCT scans were unsatisfactory, or a suspicion of regional or distant relapse on MSCT scans appeared. Local and distant control were defined according to RECIST criteria [44]. Acute and late toxicity was scored according to the National Cancer Institute (NCI) Common Terminology Criteria for Adverse Events (CTCAE) v4.03. 

### 2.6. Statistical Analysis

Primary end points were overall survival (OS) and freedom from local progression or local control (FFLP/LC), and secondary end points were progression-free survival (PFS) and toxicity rate. One-year survival was calculated as a ratio of patients that survived at least one year (12 months) and all patients. FFLP was calculated from the time of diagnosis to the first finding of local progression, defined as radiological progression of the primary lesion within the PTV. PFS was calculated from the time of diagnosis to the first radiological assessment of regional or distal disease progression. Patients that did not develop disease progression were censored at the date of the last scan. FFLP, PFS, and OS rates were calculated from the time of diagnosis to death, following the Kaplan–Meier method, and the log-rank test statistic was used for univariate analysis. A significant difference was considered when *p* ≤ 0.05.

## 3. Results

All patients completed the treatment with no significant delays.

Daily treatments lasted typically between 30–60 min, mostly dependent on patient’s cooperation and ability to hold the breath and/or lie still. There were multiple beam-offs per patient. For patients in DBH there were typically 6–15 breath-holds per fraction, each lasting approximately 20–45 s and the times between them were beam-offs. For patients in FB, there were significantly more beam offs, as the transponders were in the correct treatment position typically 10–15% of the treatment time. Figure 3 presents an example of beam on-off gating during SABR for FB (A) and DBH (B) patients. A permanent beam-off occurred when the transponders permanently moved out of treatment position (according to stated acceptable deviations) and normal breathing or correct breath-hold did not restore the transponders back to treatment position. This was typically due to patient’s discomfort or fatigue (as reported by the patient) or significant bowel movement (as confirmed with CBCT). In either case, an immediate patient repositioning, or a short treatment break, and then patient repositioning were required for restoration of transponders to treatment position. Each patient repositioning was performed using CBCT.

Median follow-up was 20 months (range 5–57 months). Median OS was 24 months (range 7–57 months). Median time from diagnosis to SABR was three months (range 1–10 months). One-year FFLP was 100%, one-year OS was 90.7%, and one-year PFS was 72.2%. Thirty-five patients (64.8%) received BED_10_ ≥ 112.5 Gy (median BED_10_).

Table 5 summarizes the results for FFLP, PFS, and OS from the diagnosis, and OS from the treatment (OSt). In Figure 4, an actuarial curve for OS is shown.

Thirty-eight patients (70.4%) received systemic treatment, either gemcitabine-based or fluorouracil-leucovorin-irinotecan-oxaliplatin (FOLFIRINOX), starting before (typically 2–4 cycles) and continuing after SABR, or starting the systemic treatment after SABR. Sixteen patients (29.6%) received no systemic treatment. On actuarial analysis, patients that received chemotherapy had significantly better OS (log-rank, *p* = 0.02) (Figure 5A)

Four patients (7.4%) had radiological local disease progression (at 25, 39, 42, and 44 months, all accompanied by distal progression). Twenty-one patients (38.9%) had a radiological local regression. The remaining 29 patients (53.7%) had radiological stable local disease. On actuarial analysis, patients with local regression had significantly better OS (log-rank, *p* = 0.05) compared to patients with locally stable disease and local progression (those two subgroups were merged due to small proportion of patients with local progression) (Figure 5B).

Median PFS was 18 months (range 7–44 months). Thirty-three patients (61.1%) had systemic progression of the disease, and 21 patients (38.9%) had systemic stable disease. On actuarial analysis patients that had systemic stable disease had better OS (log-rank, *p* = 0.001) (Figure 5C).

Median tumor volume (GTV) was 35.8 cm^3^ (range 6.4—126.1 cm^3^) and median PTV was 56.2 cm^3^ (range 10.4–161.1 cm^3^). On actuarial analysis, there was a positive impact of smaller tumour volume on OS, as the patients with tumor volume below the median had statistically better OS (log-rank, *p* = 0.02). (Figure 5D).

Thirty-one (57.4%) patients had grade 1 or 2 acute toxicities: nausea, fatigue, and abdominal spasm or pain that were successfully treated with symptomatic treatment (proton pump inhibitors, antiemetics and spasmolytics). Remaining patients reported no toxicities. Five patients (9.3%) had G2 late toxicity (abdominal spasm or pain, and/or gastroesophageal reflux), developed six months or later after SABR, that was successfully treated with symptomatic treatment. Six patients were treated with a single fraction. Four of them (66.7%) had grade 2 acute toxicities and three (50%) had grade 2 late toxicities. No acute or late G3 toxicity (ulcer, bleeding from gastrointestinal track or perforations) was reported.

Twenty-five patients (46.3%) were alive at the time of analysis. Median follow-up in this subgroup was 26 months (range 16–57 months) and median OS was 26 months (range 16–57 months). Twenty-one patients (84.0%) from this subgroup received systemic treatment, and 19 patients (76.0%) received BED_10_ ≥ 112.5 Gy (median BED_10_).

Figure 6 represents a typical SABR treatment, and a clinical response on follow up.

## 4. Discussion

During recent years, the role of SABR as the treatment for patients diagnosed with LAPC has been more thoroughly investigated, with goals to research and confirm the hypothetical advantages of this therapy over conventional systemic therapy and/or radiochemotherapy, as well as the possible positive impact of dose escalation to LC and OS. In Table 6, a summary of the number of patients, motion management techniques, follow up, one-year LC, median OS, acute toxicities, fractionation regimes, and BED_10_ in recently published studies is shown.

As presented by Comito T, et al., SABR provided high LC and OS rates, related to the high doses used (BED_10_ = 78.8 Gy), short overall treatment time (six fractions), satisfying toxicity profile, and integration with systemic therapy. A satisfactory LC rate was shown, particularly for smaller lesions (<3.5 cm), and promising OS rates were achieved in patients treated with added systemic therapy. Moreover, authors confirmed the significant correlation between LC and OS in their study [48].

Three studies from 2019 showed the importance of dose escalation. Zhu X et al. concluded that at least BED_10_ ≥ 60 Gy might be required to achieve better treatment outcomes in pancreatic cancer [61], and Bruyenzeel AME et al. concluded that delivering BED_10_ in excess of 70 Gy has not shown to significantly improve LC [62]. Reyngold, M. et al. in their review summarized the evolution of the radiation techniques over time, from conventional to ablative, and stated that advanced organ motion management, image guidance, and adaptive planning techniques enabled the delivery of ablative doses of radiation (BED_10_ ≥ 100 Gy), and that this approach resulted in encouraging improvements in survival in several studies [63].

MR-guided radiotherapy (MRgRT) and stereotactic magnetic resonance-guided adaptive radiation therapy (SMART) with daily online adaptation using magnetic resonance-guidance on-table reoptimization, are currently investigated motion management techniques. Fractionation regimens, BED_10_ and reported clinical outcomes from published studies using this adaptive planning approach are presented in Table 6. Moreover, the ongoing prospective phase II study (NCT03621644) of SMART for borderline and LAPC aims to present the clinical outcomes with this approach, i.e., OS, PFS, gastrointestinal toxicities, and patients’ quality of life. The addition of magnetic resonance-guided techniques to SABR potentially allows dose escalation and the conversion of unresectable tumours to operable cases [64]. Online adaptive approach is very appealing. However, it is not applicable with the Calypso^®^ system, but Calypso^®^ technology could be practiced widely as a motion management upgrade to existing gantry-based linacs equipped for SBRT, with on-board CBCTs.

In our study, we aimed to escalate the BED_10_ even further than previously reported, as NCCN Guidelines [22] permit dose escalation, as long as OARs constraints are respected. In our study, 64.8% of patients received BED_10_ ≥ 112.5 Gy. We were hoping to consequently increase LC and OS, and to decrease the toxicity rate, or at least keep it within reported rates. There is still virtually no published data regarding Calypso^®^ tracking system as a motion management for LAPC. The system provided precise and continuous tumour motion tracking of all possible tumor movements during SABR, with submillimetre accuracy, allowing the PTV-CTV margin reduction down to 3 millimetres, consequently providing better OARs spare that enabled safer dose escalation to the tumor. More heterogeneous dose distributions were also allowed, which might have been potentially beneficial in treating central, hypoxic and radioresistant areas inside the lesion, while allowing a steeper dose falloff outside, which could also contribute to OARs spare. In the study from Hoyer et al. (2005), the importance of motion management was shown, as they applied BED_10_ = 112.5 Gy (in three fractions) to the primary pancreatic lesion using large PTV-CTV margins and patient immobilization solely. Rather poor outcomes were reported, namely a median OS of 5.7 months, one-year survival of 5%, and 44% of grade 3–4 toxicities [65].

During the first year of the treatments, our median BED_10_ was 85.5 Gy, but after one year of gathered experience and eleven patients (20.4%) treated, our median BED_10_ for new patients was cautiously and gradually raised to 112.5 Gy, and six patients (11.1%) with particularly favourable anatomy were treated with a BED_10_ = 129 Gy in a single fraction. The patients treated with a single fraction were among the last treated and had consequently shorter follow up time, so the role of such dose escalation in a prognosis of the outcomes for the patients with LAPC remains to be further investigated. Moreover, our PTV-CTV margins’ calculations were based on the data from two published papers on PTV-CTV margins for SABR using the Calypso^®^ system for prostatic cancer [66,67], as they were, to our best knowledge, the only compatible references at the time.

Our study further confirms the known fact that the distant relapse remains the main pattern of failure in patients with LAPC, as 61.1% of our patients had regional and/or distant relapse. This is in accordance with previously reported results and with the assumption of the presence of regional and/or distant micrometastasis in apparently localized cases.

The considerable proportion of patients in our study (46.3%) were alive at the time of the analysis, with the median OS of 26 months, and this outcome is rather comparable with reported medians of OS for radically resected patients following adjuvant therapy, which is 20 to 28 months [68]. Furthermore, a larger proportion of patients in this subgroup received both systemic therapy and dose escalation (BED_10_ ≥ 112.5 Gy), compared to all patients, which could lead to the potential conclusion that more aggressive local treatment with added systemic treatment could yield better clinical outcomes. This potential conclusion could also be supported by the following findings in our study: patients with the regression of primary tumor on diagnostic imaging during follow up, as well as patients that received systemic treatment had significantly better OS, respectively. We found a similar impact of smaller tumor volume and application of systemic therapy on OS as in [47]. Optimal regimens and settings of local and systemic treatment combined remain to be investigated.

The toxicity profile of SABR in our study was very acceptable, as no patient experienced acute or late grade 3 toxicities. According to published literature, the authors predominantly consider that acute and late grade ≥ 3 toxicities are strongly related to a single fraction treatment. Although the acute and late grade 2 toxicities in our study did appear predominantly among the patients treated with a single fraction, it is our impression that general absence of grade 3 toxicities in our study could actually have been related to precise motion management and OARs sparing.

The Beacon^®^ transponders implantation represented, in our experience, no additional risk for the patients, as there were no complications noticed during the procedure or reported after. The Calypso^®^ system also eliminated any need for immobilization and/or motion mitigation during SABR, making the treatment more comfortable for the patients, and offering a significant treatment time shortening (64.8% of our patients were treated in one or three fractions), allowing for faster workflow as well as better patient compliance and quality of life. The notable disadvantage was a delay in treatment, due to the time required for the implanted transponders’ in-site stabilization. No significant migration (≥2 mm in any direction) of any transponder from its initial position or loss were noticed.

The authors are aware of all disadvantages of this study, based on its retrospective nature. However, we consider our results potentially valuable and encouraging, as well as helpful to assess the feasibility and design of future prospective studies on this topic.

## 5. Conclusions

SABR combined with the Calypso^®^ intrafractional, fiducial-based motion management presented in our study as an effective and safe local treatment for LAPC, as potentially improved local control and overall survival with very acceptable toxicities were shown. Our results indicated that the Calypso^®^ system provided a precise tumor motion management that enabled effective OARs sparing, consequently allowing the significant dose escalation with a high dose heterogeneity inside the lesion and steep dose falloff outside, which all could lead to possible improvements of clinical outcomes for the patients. This local therapeutic option could be considered potentially effective as a single treatment, and even more effective as a part of the multimodality treatment for this disease.

Future prospective studies and trials are needed to evaluate the role of dose escalation in the improvement of clinical outcomes for these patients, as well as the optimal integration of SABR with systemic treatment.

## Figures and Tables

**Figure 1 cancers-14-02688-f001:**
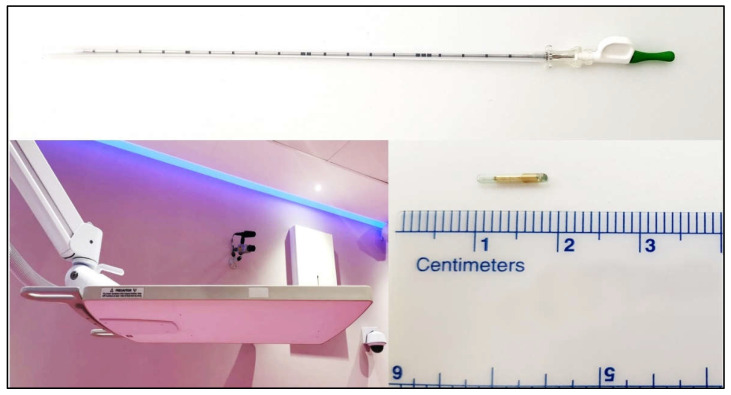
Implantation needle (**top**); electromagnetic array (**bottom left**); Beacon^®^ transponder (**bottom right**).

**Figure 2 cancers-14-02688-f002:**
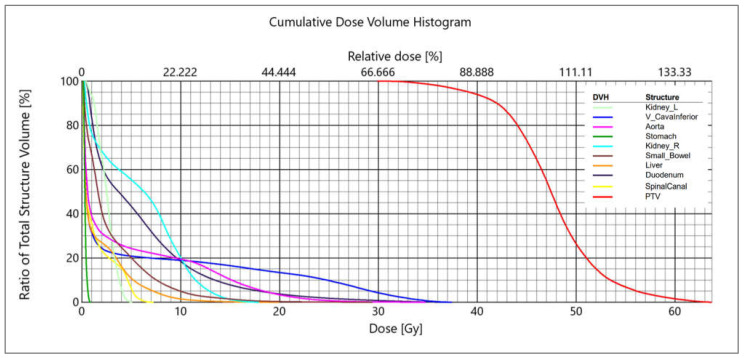
Example of DVH—dose was prescribed as 45 Gy in 3 fractions to the PTV—98% of PTV volume was covered by a dose of 36 Gy (80% of the prescribed dose), the median dose to the PTV was 47.5 Gy, and the maximum dose was 63.7 Gy (141.6% of the prescribed dose).

**Figure 3 cancers-14-02688-f003:**
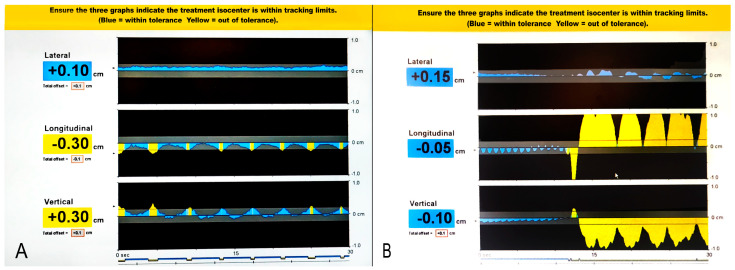
An example of beam on-off gating during SABR for FB (A) and DBH (B) patients—blue areas under the curve denote the transponders within the gating windows (presented as a dark grey areas), and yellow areas under the curve denote the transponders exceeding the gating windows.

**Figure 4 cancers-14-02688-f004:**
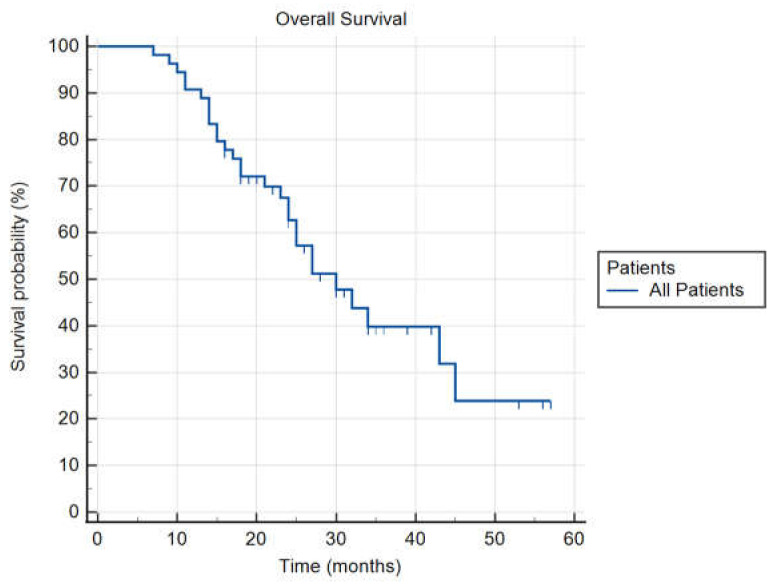
Actuarial curve of Overall Survival for all patients.

**Figure 5 cancers-14-02688-f005:**
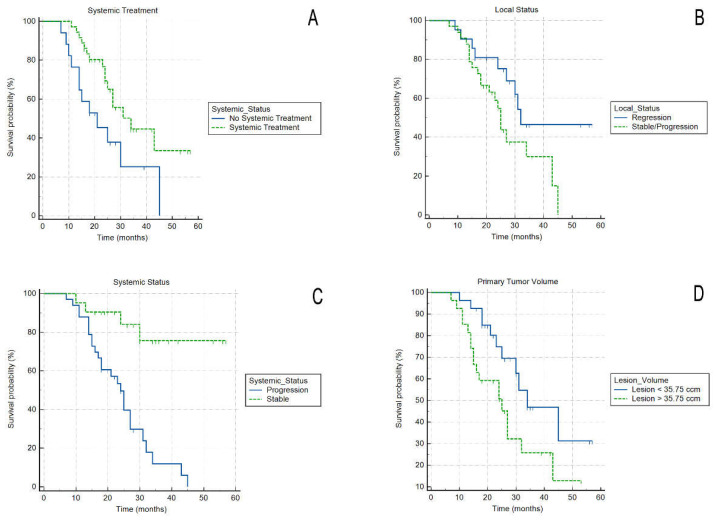
Actuarial curves of survival for: patients receiving/not receiving systemic treatment (**A**); patients with or without local tumor regression after the treatment (**B**); patients with or without systemic progression after the treatment (**C**); and for patients with tumour smaller or larger than median volume (**D**).

**Figure 6 cancers-14-02688-f006:**
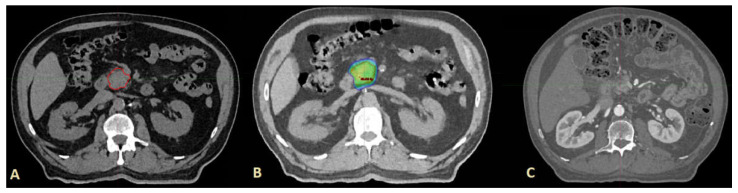
An example of treatment: (**A**) Gross tumour volume pre-SABR (contoured red); (**B**) dose distribution with colour wash set at 80% to 130% of prescription dose; (**C**) treatment response (complete tumour regression) at 21 months follow-up.

**Table 1 cancers-14-02688-t001:** Patients’ Characteristics.

Patients’ Number	54
Mean age in years (range)	67 (45–87)
Sex (M:F)	30:24
Primary site	
Head	41 (76%)
Body/tail	13 (24%)
Systemic treatment	
Gemcitabine-based	18 (33%)
FOLFIRINOX	20 (37%)
No systemic treatment	16 (30%)
Median CTV	35.8 cm^3^ (range 6.4–126.1 cm^3^)
Median PTV	56.2 cm^3^ (range 10.4–161.6 cm^3^)

Abbreviations: CTV—clinical target volume; PTV—planning target volume.

**Table 2 cancers-14-02688-t002:** SABR protocol and dose delivery techniques.

Fifty-Four Patients Enrolled ↓ Initial Test for Deep Breath-Hold ↓ ↓	
**Treatment in deep breath-hold** **15 patients (27.8%)**	**Treatment in free breathing** **39 patients (72.2%)**
Respiratory phase gating—“Beam on-off” technique: Therapeutic position of the tumor is in a deep breath-hold Dose delivery using Volumetric Arc Therapy (RapidArc^®^)	Respiratory phase gating—“Beam on-off” technique: Therapeutic position of the tumor is in a single, late exhale phase on 4D CT Dose delivery using intensity modulated radiotherapy—IMRT

**Table 3 cancers-14-02688-t003:** Dose regimens and corresponding BED_10_.

Regimen	BED_10_	Number of Patients
**5 × 9 Gy**	85.5 Gy	19 (35%)
**3 × 15 Gy**	112.5 Gy	29 (54%)
**1 × 32 Gy**	129 Gy	6 (11%)

**Table 4 cancers-14-02688-t004:** Dose-Volume Constraints.

Organs at Risk	One Fraction	Three Fractions	Five Fractions
**Primary OAR (Stomach,** **Duodenum,** **Small Bowel)**	D_max_ (0.03 cm^3^) < 23 Gy V_(20 Gy)_ < 3.3 cm^3^ V_(15 Gy)_ < 9.1 cm^3^	V_(31.4 Gy)_ < 1cm^3^ V_(23.3 Gy)_ < 5 cm^3^ V_(16.1 Gy)_ < 10 cm^3^	V_(42 Gy)_ < 1 cm^3^ V_(25.4 Gy)_ < 5 cm^3^ V_(17.6 Gy)_ < 10 cm^3^
**Liver**	V_(9.1 Gy)_ < 700 cm^3^	V _(17 Gy)_ < 700 cm^3^	V_(21 Gy)_ < 700 cm^3^
**Great Vessels**	D_max_ < 37 Gy	D_max_ < 45 Gy	D_max_ < 53 Gy
**Spinal Cord**	D_max_ < 14 Gy	D_max_ < 22 Gy	D_max_ < 30 Gy
**Kidneys**	V_(8.4 Gy)_ < 200 cm^3^	V_(14.4 Gy)_ < 200 cm^3^	V_(17.5 Gy)_ < 200 cm^3^

**Table 5 cancers-14-02688-t005:** Summary of the Actuarial Analysis for FFLP, PFS, OS and OSt.

End Points	Median	1 Year
FFLP	40.5 months *	100%
PFS	18 months (95% CI: 14.3 to 19.2)	72.2%
OS	24 months (95% CI: 21.9 to 28.9)	90.7%
OSt	21 months (95% CI: 18.6 to 24.4)	81.5%

* Four cases of local failure at 25, 39, 42, and 44 months. Abbreviations: CI, confidence interval; FFLP, freedom from local progression; OS, overall survival; OSt, overall survival from the treatment; PFS, progression-free survival.

**Table 6 cancers-14-02688-t006:** Summary of number of patients, motion management techniques, follow up, 1-year local control, median overall survival, toxicities, fractionation regimes and BED_10_ in published studies.

Study (Chronologically)	Number of Patients	Motion Management	Follow Up (Months)	One-Year Local Control	Median Overall Survival (Months)	Toxicity Grade ≥ 3	Fractionation Regimens	BED_10_ (Gy)
Mahadevan A et al. (2011) [17]	39	-	-	85%	20	41%	3 × 8–12 Gy	43.2–79.2
Rwigema J et al. (2011) [45]	71	-	12.7	64.8%	10.3	0%	1 × 18–25 Gy	50.4–87.5
Gurka M et al. (2013) [46]	11	Fiducial-based	-	40%	12.2	0%	5 × 5 Gy	37.5
Chuong MD et al. (2013) [19]	73	-	10.5	81%	15	0%	5 × 5–10 Gy	37.5–100
Tozzi A et al. (2013) [20]	30	Abdominal compression	11	85%	19.5	20%	6 × 6–7.5 Gy	48–78.8
Herman JM et al. (2015) [47]	49	Fiducial-based	13.9	78%	13.9	0%	5 × 6.6 Gy	54.8
Moningi S et al. (2015) [48]	88	Fiducial-based	14.5	-	18.4	3.4%	5 × 5–6.6 Gy	37.5–54.8
Comito T et al. (2017) [49]	45	Abdominal compression	13.5	90%	19	0%	6 × 7.5 Gy	78.8
Seo J et al. (2017) [50]	79	Respiratory gating	11	96%	16	4%	4 × 6–8 Gy	38.4–57.6
Zaorsky NG et al. (2017) [51]	520 (meta-analysis)	-	9.1	66%	13.3	0%	5 × 6 Gy	48
Mazzola R et al. (2018) [52]	33	-	18	81%	-	0%	6 × 6 −7.5 Gy	48–78.8
Herkens HD et al. (2018) [53]	20	MRgRT + implanted fiducials + abdominal compression	-	-	8.5	0%	3 × 8 Gy	43.2
Park HH et al. (2019) [54]	95	-	15	-	17.3	1%	24–36 Gy/5–6 Gy per fx.	38.4–48
Rudra S et al. (2019) [55]	44	MRgRT	17	77% (2-y)	-	7%	25 × 2 Gy to 5 × 10.4 Gy	60–106
Chuong MD et al. (2020) [56]	35	SMART	10.3	87.8%	9.8	2.9%	5 × 10 Gy	100
Hassanzadeh C et al.(2020) [57]	44	SMART	16	68.2%	15.7	4.6%	5 × 10 Gy	100
Placidi L et al. (2020) [58]	8	MRgRT	13	75%	-	0%	5 × 6–8 Gy	48–72
Hall WA et al. (2021) [59]	Review of 300 manuscripts	MRgRT	-	77–87.8%	9.8–15.7	2.9–4.6%	25 × 2 Gy to 5 × 10 Gy	60–100
Michalet M et al. (2022) [60]	30	SMART	10.6	70%	14.1	0%	5 × 6–10 Gy	48–100
Current Study	54	Calypso Fiducial-based	20	100%	24	0%	5 × 9 Gy 3 × 15 Gy 1 × 32 Gy	112.5 (85.5–129)

Abbreviations: SMART, Stereotactic MRI-guided Adaptive Radiotherapy; MRgRT, MR-guided radiotherapy.

## Data Availability

Research data are stored in an institutional repository and will be shared upon request to the corresponding author.

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
