# Peer review of "Stereotactic Ablative Radiotherapy Using CALYPSO® Extracranial Tracking for Intrafractional Tumor Motion Management—A New Potential Local Treatment for Unresectable Locally Advanced Pancreatic Cancer? Results from a Retrospective Study"

_cancers, 2022, doi:10.3390/cancers14112688_

Round 1

Reviewer 1 Report

Major Comments:

The manuscript deals with an important disease site in radiation therapy, where clinical outcome improvements are currently under investigation, but which demonstrated poor clinical outcomes using traditional radiation therapy regimes in the past. It was only when adaptive therapy together with real-time tracking was used for this indication that researchers demonstrated significant improvements in clinical outcomes.

The authors present their impressive clinical results in the manuscript and compare those with published results. This is an important contribution to the field and should be published after the authors did improve their manuscript based on found deficits.

Recent publications addressing clinical outcomes are missing and need to be added. Major contributions such as https://www.frontiersin.org/articles/10.3389/fonc.2022.842402/full, https://www.sciencedirect.com/science/article/pii/S2405632420300159, https://onlinelibrary.wiley.com/doi/epdf/10.1002/cam4.2100, and the review: https://www.frontiersin.org/articles/10.3389/fonc.2021.628155/full are missing. Please note that this is not a full literature search and the authors are encouraged to complete this list. The authors are encouraged also to mention the ongoing trial NCT03621644 on Stereotactic MRI-guided On-table Adaptive Radiation Therapy (SMART) for Locally Advanced Pancreatic Cancer.

In addition, the manuscript is focusing on intra-fraction motion management but is not using an adaptive planning approach compared to other studies referenced. The authors are encouraged to discuss this aspect in their manuscript and if an adaptive approach may be beneficial given their clinical experience and if such an approach could even further improve the excellent clinical outcome presented in this study. This is especially important as the authors use a non-MRI motion management technology for intra-fraction motion management and such a technology could be practiced widely using standard radiation therapy devices.

The authors are encouraged to compare their dose and fractionation scheme and dose prescription including BED10 to the ones used in studies mentioned in Tab. 6.

Finally, the authors are encouraged to clean up the manuscript according to found minor deficits listed below prior to publication.

Minor Comments:

  • Line 19: represent a particular instead of representing particular
  • Line 39: presented as an effective and safe treatment? Please check grammar as there is a word missing after safe.
  • Line 81: Please add a reference to this ASTRO Clinical Practice Guideline
  • Line 83: eventual surgery – to enable? Please check
  • Line 84: the treatment volume instead of a treatment volume
  • Line 90: intensity instead of intense
  • Lines 96-100: Here various motion management strategies are quoted. Please add references as being appropriate.
  • Line 99: breath-hold (DBH) patients – please check grammar. You may consider deleting patients.
  • Line 114: Calypso Extracranial Tracking – As there is no Calypso Intracranial Tracking option, consider deleting Extracranial (see also Lines 152, 411)
  • Line 131: institutional instead of Institutional also in Line 135
  • Table 1: mean age in years instead of years
  • Line 136: tumor instead of Tumor
  • Line 146: informed instead of Informed
  • Line 148: institution instead of Institution (see also line 176)
  • Line 154: approved for instead of approved for a
  • Line 170: shows instead of presents
  • Line 171: instead of “the figure is taken from our previous paper” add the reference of your previous paper
  • Fig 2: Indicate which curve belongs to which structure.
  • Line 243: RTOG recommendation is linked to reference 42, which is an AAPM recommendation. Please check if this is correct.
  • Line 249 in FB instead of in a FB.
  • Line 255-256: this statement requires more elaboration and evidence of adverse dosimetric effects when using gated VMAT. Consider adding a reference supporting this non-scientific statement.
  • Line 260: weight instead of primacy
  • Line 264: Explain why it is OK to accept up to 20 degrees of rotation
  • Line 289: coaching instead of couching
  • Line 298: day clinic instead of daily clinic
  • Fig 4: to be consistent with Fig. 5 the y-axis should start at 0 and not 20%
  • Table 6: add more recent publications. Consider also adding information on whether these studies were using an adaptive concept. Add also the information on the motion management techniques used in these studies. Were these studies all using SABR?
  • Line 426: new patients instead of further patients?
  • Line 431: confirms the known instead of confirms known
  • Line 446: as in [48] instead of as the Italian group of authors [48]
  • Line 454: OAR sparing instead of OARs spare
  • Line 463: initial position instead of initial spot

Reviewer 2 Report

The authors presented an interesting study with 45 patients treated with SABR for LAPC using the Calypso system. To our best knowledge, this is the first reported work with this type of tracking system for LAPC. Large part of the work is about the description of the clinical workflow and the clinical outcomes.

Although potentially very interesting as motion tracking study for pancreatic cancer during radiotherapy – supposing this as the focus of the work, given the title of the paper - the motion analysis is very limited, as no data about the pancreas motion during treatment was provided. On the other side, large part of the text is about the clinical outcomes of the treatment. The authors compared this data with the results of analogous studies in literature, without emphasizing  enough the possible novelty of their work.

The paper is very well written in English, but it should be better organized. Several part relative to Material, Results and comments were misplaced in the draft. 

Major revision

L. 175. Transponder implant in the pancreas (which is not superficial) is a non standard application for Calypso. Could you please give more details of this surgical procedure? Any reported issue?

l.196: which guidelines did you use for target delineation and contouring? For FB, do you delineate ITV? And margins in general? Please add important details about target delineation and margins (with respect to motion management as well) here.

l. 244 and table 3. Given the large literature for toxicity with photons and IORT, do you use any constraint for the portal vein?

l. 267 – 279: this part should go in the result, in particular as a first part of a motion analysis which is at the moment missing. Respiratory motion analysis? Average displacement of the pancreas? Rigid displacement or deformation of the pancreas? Could you estimate for how long the target was within the margins? Bowel movement were present?

l. 277: patient discomfort or bowel movement. How could you decide which is the correct one if the patient was not immobilized with a mask? In a time period of 30 minutes to 1 hour, OARs could change their position and shape with respect to the acquired position with the CBCT. How to detect this? Were the transponder position enough in your opinion?

l. 284: patient preparation is reported after the description of the treatments. Please re-organize your material.

l. 362. Any reported portal vein toxicity?

l. 414. PTV-CTV margin reduction down to 3 mm. It was not reported in the text the used treatment margin. On which analysis was this reduction carried out? Usually an analysis about systematic and random errors (see van Herk) leads to a margin reduction. Could you describe here better?

l.457. the percentage of beam-off time was significant (>80% as reported before). Please comment about that relative to the lack of an immobilization mask (might it reduce the variation due to patient breathing? Might it reduce the patient placement on the cushion during the treatment?). Usually longer treatment sessions reduce treatment delivery quality and patent comfort.

Minor revision

l. 80 – 92: I suggest to report only the info necessary to explain the rest of the paper, not an excerpt from the protocol.

l. 101 – 110: please rephrase this part underlying what is interesting and not reporting several sentences from different papers.

l. 114: “up to now” change in “to our best knowledge”

l. 131: Istitutional ethics committee…please add protocol code.

l. 200- 230. Little confusing please rephrase

l. 264. These thresholds were set relative to a single transponder? To the sum of the transponders?

Round 2

Reviewer 2 Report

I commented some of your previous answers. Find it in blue.

  1. 244 and table 3. Given the large literature for toxicity with photons and IORT, do you use any constraint for the portal vein?
  • We did not use any constraints for the portal vein

ESTRO protocol [26] recommended Dmax < 45 Gy (for 3 fr) and Dmax < 53 Gy (for 5 fr)

  1. 277: patient discomfort or bowel movement. How could you decide which is the correct one if the patient was not immobilized with a mask? In a time period of 30 minutes to 1 hour, OARs could change their position and shape with respect to the acquired position with the CBCT. How to detect this? Were the transponder position enough in your opinion?
  • Patient’s discomfort or fatigue were usually reported by the patient – in such cases we let the patient to put the hands down and to rest for a while on a treatment table, or we did a short treatment break - the patient was instructed to have a walk and relax for a while, and then we continued the treatment

Do you acquire CBCT after each “walk”? ESTRO protocol [26] recommended a new CBCT for treatment longer than 15 min. Please comment on that

  • when we assumed that transponders moved permanently due to OARs’ (typically bowel) significant movements/displacements, the CBCT was performed and if the OARs’ significant movement was confirmed, we did a short treatment break and the patient was instructed to have a walk and relax for a while, and then we continued the treatment
  • In rare cases when none of above resulted in restoration of the transponders to treatment position, we scheduled the patient for the next day with instructions on diet and antiflatulent and mild sedative medication, as needed.
  • Such cases were rare, due to very strict instructions (written and oral) on diet and antiflatulent medication that every patient was given on a day of transponder implantation
  • Either in FB or in DBH patients, the lesion was treated in a single breathing phase, so as long as the transponders were in a correct treatment position, we assumed that there were no significant OARs’ movements.

This is a very strong assumption that should be justified and commented. Transponders provide the target position,  the correct position of the transponders do not imply a correct position of the OARs. Moreover, the acceptance window for each of the transponders position is 2 mm; when all transponders are out of position of 2 mm, this could potentially translate to a 3D vector of 3.5 mm. Following this reasoning, the target could be out of position of 3.5 mm and the bowel could be in that position, receiving the prescription dose in that fraction. Given that you are not using PRV for the primary OARs and you do not monitor the OARs position along the treatment (assuming that the treatment last between 30 and 60 minutes, as you stated), you should justified this assumption based on a worst case scenario analysis or quantitative data.

  1. 414. PTV-CTV margin reduction down to 3 mm. It was not reported in the text the used treatment margin. On which analysis was this reduction carried out? Usually an analysis about systematic and random errors (see van Herk) leads to a margin reduction. Could you describe here better?
  • The analysis was carried out and appropriate PTV-CTV margin was calculated by our medical physicists, using van Herk’s formula (2.5Σ + 0.7σ)
  • The explanation is added to the chapter “4. Stereotactic Ablative Radiotherapy“

Please provide more details about that. The acceptance window for each of the transponders position is already 2 mm. Target motion is described by a 4DCT during planning, but then another system (the transponders) are used for motion monitoring and to gate treatment. What is the uncertainty between the end-exhale phase that you use for planning (as described by the 4DCT) and the one you used for the treatment (as described by the transponders)? In the cited paper by your group [40] you wrote: “The differences between the Calypso® and 4D-CT median recordings of maximal tumor excursion were in the CC, AP and LR directions 10 mm, 5 mm and 2 mm, respectively”. Then you should consider also patient setup and imaging uncertainties. In my opinion only 1 mm (to take into consideration that the transponders could be 2 mm out of position) is small. Could you please specify the details of the analysis you made to calculate a PTV margins of 3 mm?

l.281 Initial positioning was performed in FB or DBH (depending on treatment modality), before each fraction, by registering cone beam CT (CBCT) imaging with the planning MSCT, using soft tissue and bony anatomy and the transponders, with weight given to the transponders’ position for registering.

For the patients treated in DBH this is clear. But for patients treated in FB, you are registering the CBCT image acquired in FB, with the 4DCT reconstructed at the end exhale phase. How could you register these two datasets acquired in different breathing conditions?

Figure 2. Provide the DVH for stomach, duodenum and small bowel (primary OAR, as defined by the authors) as well.

Round 3

Reviewer 2 Report

I thank the authors for their extensive answers. The manunscript has been substantially improved from my point of view.